# Current Management of Generalized Convulsive Status Epilepticus in Children

**DOI:** 10.3390/children9101586

**Published:** 2022-10-20

**Authors:** Štefania Aulická

**Affiliations:** Department of Pediatric Neurology, University Hospital and Medical Faculty of Masaryk University, 61200 Brno, Czech Republic; stefania.aulicka@gmail.com

**Keywords:** generalized convulsive status epilepticus, management, propofol infuse syndrome, children, burst-suppression pattern

## Abstract

Generalized convulsive status epilepticus (GCSE) in pediatric patients is an emergency condition with high morbidity and mortality and potentially irreversible brain damage, leading to cognitive deterioration, psychomotor retardation, chronic epilepsy with recurring seizures, and other complications. Treatment must be initiated in the impending GCSE phase, within five minutes of the onset of a generalized convulsive seizure. Early initiation of treatment and adequate therapy is a prerequisite for a good patient outcome.

## 1. Introduction

Generalized convulsive status epilepticus (GCSE) is the most common emergent condition in children in neurology. This is a serious, often life-threatening condition, requiring urgent therapeutic intervention.

### 1.1. Epidemiology

The estimated incidence of status epilepticus (SE) in childhood is 20 per 100,000 per year. Neonatal age is the riskiest period for the development of SE. Increased seizure alertness for this age group is necessary due to the unfinished maturation of the brain. Febrile seizures are the most common cause of SE. Before their first onset of SE, 60% of children are neurologically healthy. In children with epilepsy, 10–20% experience SE at least once time. In 12% of children, SE is the first manifestation of epilepsy [1].

### 1.2. Pathophysiology and Etiopathogenesis

From a pathophysiological point of view, SE is defined as a condition caused by the failure of the mechanisms responsible for the cessation of seizure and/or the development of mechanisms leading to abnormal prolongation of seizure. Inhibitory GABAergic (GABA A) receptors and excitatory glutamatergic N-methyl-D-aspartate (NMDA) receptors play a decisive role in the development of SE.

Glutamate is the main excitatory neurotransmitter in the brain. Other excitatory neurotransmitters are aspartate and acetylcholine. Gamma-amino-butyric acid (GABA) is the main inhibitory neurotransmitter in the brain. The potentiation of GABA-mediated neuronal activity is the essence of the mechanism of action of benzodiazepines, barbiturates, and propofol that bind to GABA A receptors.

Continued seizure activity leads to the internalization of inhibitory GABA A receptors and their loss on the postsynaptic membrane, while the number of functional excitatory NMDA receptors increases.

The decrease in GABA A receptors on the membrane explains the rapid loss of the effect of benzodiazepine.

These changes, which occur in the horizon of minutes to hours, maintain a state of increased excitability.

It is believed that neuronal losses occur during SE. In repeated epileptic states, neuronal losses accumulate and lead to cognitive impairment [2].

### 1.3. Definition

SE is (from a clinical point of view) defined as an epileptic seizure lasting more than five minutes. Generalized convulsive seizures are convulsions lasting more than five minutes with unconsciousness or repeated seizures between which the patient does not regain consciousness. Within five minutes, adequate treatment leading to the termination of the seizure must be initiated.

In 2015, the International League Against Epilepsy (ILAE) Classification and Terminology Commission established a new definition of SE. This definition emphasizes two critical points (t1, t2):►t1—The time at which spontaneous termination of epileptic activity is unlikely and adequate therapy should be initiated at this point at the latest;►t2—The time beyond which continued epileptic activity can cause long-term pathological changes in the brain (alteration of neuronal network function, neuronal damage, neuronal death).

The time aspect is best defined in generalized CSE: the time point t1 = 5 min, t2 = 30 min. Adequate treatment must be initiated during the initial 5-min time window. If a generalized convulsive seizure is not under control within 5 min, there is a risk of its continuation, with the risk of irreversible brain damage after 30 min.

Delayed initiation of treatment is associated with a worse response to treatment. For individual types of SE, these time limits vary and, especially with non-convulsive states, they are not clear. For example, in focal SE with alteration of consciousness, t1 = 10 min and t2 = 60 min; in the absence of SE, t1 = 10–15 min and t2 has not yet been defined [3].

### 1.4. Classification

In clinical practice, it is possible to use a simplified version of the classification by previous authors [3]:►Focal SE without impaired consciousness—continuous or recurrent focal motor or sensory seizure activity without alteration of consciousness;►Focal SE with impaired consciousness—continuous or recurrent focal motor (automatisms) or sensory seizure activity with altered consciousness;►Generalized convulsive SE—tonic-clonic, tonic, or clonic convulsions always associated with unconsciousness;►Absence of SE—generalized seizure activity characterized by alteration of consciousness, but not always unconsciousness;►Myoclonic SE—continuous myoclonia can occur in diseases of a wide spectrum of severity, from benign epileptic syndromes (juvenile myoclonic epilepsy) to conditions with poor prognosis (postanoxic encephalopathy).

### 1.5. Notes on Semiological Classification:

►Generalized convulsive seizures are divided into primarily generalized and focal seizures transitioning into a bilateral tonic-clonic seizure (focal bilateral tonic-clonic seizure). These arise from the generalization of the seizure activity of the epileptogenic focus. Clinically, the two types usually cannot be distinguished (generalization usually occurs so quickly that no focal symptoms manifest themselves);►Absence SE and focal SE (with or without impaired consciousness) without motor symptomatology are denoted by the term non-convulsive SE (also subclinical SE). Non-convulsive SE can only be diagnosed electroencephalographically;►Non-convulsive SE may also be followed by untreated or inadequately treated generalized convulsive SE. In this case, it is called subtle SE. Patients tend to be comatose, possibly with inconspicuous motor symptoms, such as minor generalized myoclonia or twitching of the eyeballs. Subtle SE is the non-convulsive epileptic state with the worst prognosis. Other authors have excluded subtle SE from the group of non-convulsive SEs.

### 1.6. Etiopathogenesis

Similar to an isolated seizure, SE can be a manifestation of acute brain involvement or a symptom of epilepsy [4].

The causes of SE in childhood include:►Hypoxic-ischemic encephalopathy;►Intracerebral hemorrhage;►Neuroinfection (pre-, peri-, post-natal);►Cortical dysplasia and CNS malformations (disorders of migration, segmentation, myelination, synaptogenesis, etc.);►Metabolic causes (hypocalcemia, hypoglycemia, hypomagnesemia, hyponatremia);►Febrile status epilepticus (neuroinfection must be excluded);►Some genetic syndromes, such as Dravet syndrome and Angelman syndrome, can be manifested by recurrent attacks of SE;►Autoimmune encephalitis: Rasmussen’s encephalitis, new onset refractory status epilepticus (NORSE), febrile infection-related epileptic syndrome (FIRES), NMDA encephalitis;►Rarely, metabolic epileptic encephalopathies (1–2%).

In recent years, new onset refractory status epilepticus (NORSE) has been at the forefront of attention. The pathogenesis is unclear. Some cases may reflect autoimmune or paraneoplastic encephalitis caused by antibodies to synaptic proteins (e.g., NMDA receptors). Immunomodulatory therapy (steroids, IVIG, plasmapheresis) is the treatment of choice in this case. A subtype of NORSE, febrile infection-related epilepsy syndrome (FIRES), is a refractory SE preceded by a febrile infectious disease.

### 1.7. Diagnosis of the Etiology

In patients with CSE, the etiology is examined; the examination should take place in parallel with the initiation of general SE therapy (Table 1 here).

Findings on MR of the brain in SE: cerebral edema (based on a combination of cytotoxic and vasogenic edema), post-contrast leptomeningeal saturation (on the basis of a broken blood–brain barrier in SE), hyperperfusion of epileptogenic areas. Focal SE often include focal cerebral edema, loss of differentiation of the gray and white matter of the brain, often focal cortical dysplasia [5].

## 2. Management of the Treatment of Generalized Convulsive Status Epilepticus (GCSE)

### 2.1. Treatment in the Individual Stages of GCSE

The times of the individual stages are not uniform in the literature; the distribution below is based on the recommendations of the American Epilepsy Society [6].

### 2.2. Stages of GCSE

►Stage 1 (early GCSE; 5–20 min): initiation of general therapy, monitoring and diagnosis. Boluses of benzodiazepine. The time from 0–5 min is referred to as impending GCSE.►Stage 2 (developed GCSE; 20–40 min): continued GCSE after administration of two benzodiazepines boluses, initiation of an infusion of a non-benzodiazepine antiepileptic drug.►Stage 3 (refractory GCSE; >40 min): continued GCSE even after administration of adequate doses of benzodiazepine and non-benzodiazepine antiseizure medication (ASM), indication either for treatment with other non-benzodiazepine ASM or directly for intravenous general anesthesia.►Stage 4 (super-refractory GCSE; >24 h): GCSE continues with general anesthesia, including situations in which there is recurrence of seizure after discontinuation or reduction of anesthesia.

Although there are recommendations for the treatment of GCSE, they cannot be rigidly applied to all patients. In the following text, the time “0” indicates the beginning of a generalized convulsive paroxysm. The indicated timing may correspond to the treatment of the patient in the hospital.

If the seizure begins outside the hospital, the start of treatment by the pre-hospital care team (basic procedures and application of first choice drugs) may be delayed by 10 to 20 min or more. Therefore, it is important for epileptic patients to start treatment, e.g., the parents of a child with epilepsy should apply rectal diazepam or intranasal or buccal midazolam.

The patient can then get to the hospital after 20–30 min or more, and the second-choice drugs are usually administered in the hospital environment. However, the pre-hospital care team often leads to the introduction to general anesthesia, intubation, and artificial lung ventilation in the inefficiency of the benzodiazepine bolus.

## 3. Algorithms of GCSE Treatment in Individual Stages of GCSE

### 3.1. Impending GCSE (0–5 Min)

►Head elevation 30°;►Oxygen therapy with O_2_ mask with reservoir;►SpO_2_ and ECG monitoring (one lead), non-invasive blood pressure measurement;►Ensuring venous entry;►Collection of capillary blood for glycaemia; in cases of hypoglycemia, application of glucose bolus intravenously (in neonates 10% glucose, in larger children up to 40% glucose);►Venous blood sampling for blood counts, biochemistry, antiepileptic levels (see above);►Temperature control in case of fever or hyperthermia;►Monitoring of vital signs continues throughout GCSE therapy;►At any time (not only in the first five minutes) during GCSE therapy, impaired oxygenation (desaturation, cyanosis) and/or ventilation (hypoventilation, apnea) should lead to the introduction of general anesthesia, intubation, and ventilation

### 3.2. Early GCSE (5–20 Min)

First choice drugs (benzodiazepines, administered as an IV injection):

Diazepam * 0.15–0.2 mg/kg IV (max. 10 mg/dose) [7]. If convulsions persist after five minutes, the dose can be repeated. Possible administration of the third dose of benzodiazepine (no earlier than after another five minutes) may be considered if ventilation and oxygenation are not altered.

Notes: According to EpiStop recommendations, diazepam intravenously is administered at a dose of 0.5 mg per kg in children under three years of age and 0.3 mg per kg in children over three years of age [7].

An alternative to diazepam may be clonazepam. According to the EpiStop recommendations, the intravenous clonazepam dosage is approximately one tenth of the diazepam dose—i.e., in children under three years of age, 0.05 mg per kg, and 0.03 mg per kg in children over three years of age. Benzodiazepine therapy can be started as early as in the first five minutes. In the U.S., the first-choice drug is lorazepam.

If there is no venous entry, midazolam should be introduced intramuscularly, intranasally or buccally at 0.2 mg per kg (max. 10 mg per dose) [8]. Diazepam should be administered rectally at 0.2–0.5 mg per kg (max. 20 mg per dose) [4]. If it is not possible to secure the intravenous entry, the introduction of an intraosseous entry is indicated.

Notes: EpiStop recommendations for intramuscular, intranasal, or buccal administration of midazolam: 0.2–0.3 mg per kg for dose [7].

EpiStop recommendations for rectal application of diazepam: in children under 15 kg body weight 5 mg, over 15 kg 10 mg for dose [7].

* Diazepam is a highly lipophilic drug with rapid penetration across the blood–brain barrier. The effect of diazepam can be observed 10–20 s after administration. Due to the subsequent rapid redistribution of the drug into adipose tissue, the anticonvulsant effect of diazepam, as a rule, lasts <20 min.

If GCSE is terminated by bolus or benzodiazepine boluses, it is appropriate to consider the use of an ASM (the purpose of which is to prevent recurrence of GCSE) in patients without ASM.

### 3.3. Developed GCSE (20–40 Min)

Second choice drugs are phenytoin, valproic acid, and levetiracetam, applied as intravenous infusion; at this time there is no evidence of a preference for these drugs in terms of efficacy. Treatment with one of these drugs is started after two doses of benzodiazepines. Some algorithms recommend starting non-benzodiazepine antiepileptic therapy after the first dose of benzodiazepines. This procedure may be rational in the first manifestation of GCSE. The choice of the second-choice drug must be considered individually. Most experience is with phenytoin.

Phenytoin * 20 mg/kg (max. 1.5 g/dose) at a rate of max. 50 mg/min (e.g., 1 g phenytoin in a 20-min infusion) [6,8].

Notes:►According to EpiStop recommendations, phenytoin is administered in children under 12 years of age at a saturation dose of 20–30 mg per kg intravenously and the rate of administration is recommended to be slower than in adults (25 mg/min) [7];►Risk of hypotension and/or bradycardia (use caution in patients with heart disease—in this case, consider another second-choice medicine);►Dilute to saline (not up to 5% glucose—this would precipitate phenytoin).►Risk of seizure aggravation: Dravet syndrome, idiopathic generalized epilepsy (e.g., juvenile myoclonic epilepsy)—> use valproic acid or levetiracetam [9];►Phenytoin may not be effective for convulsions during intoxication (convulsions caused by cocaine, local anesthetics, and theophylline may even worsen);►In the U.S., fosphenytoin is used instead of phenytoin.

* Phenytoin is an antiepileptic drug with prolonged action. The effect starts after 10 to 30 min from the start of application. It follows that phenytoin therapy must be preceded by the application of a rapidly occurring antiepileptic drug, i.e., benzodiazepine.

Valproic acid 20 to 40 mg per kg (max. 3 g) in a 5- to 10-min infusion [6,8]. Note: Do not give in severe hepatopathy and/or suspected mitochondrial disorders.

Levetiracetam 40–60 mg per kg (max. 4.5 g) in a 15-min infusion (https://www.uptodate.com/contents/management-of-convulsive-status-epilepticus-in-children; Glauser et al., 2016) (accessed on 23 November 2021). Notes: Alternatively, lacosamide 10 mg per kg (max. 400 mg/dose) may be used in infusion [10,11].

The second-choice drug is also phenobarbital 15–20 mg/kg at a rate of max. 50 mg/min (e.g., 1 g phenobarbital in a 20-min infusion) [6,8].

However, compared to other second choice drugs, especially in combination with benzodiazepines, there is a risk of hypotension and, above all, respiratory depression when using it. However, it can be used as a second-choice drug for GCSE therapy in infants and especially for convulsions caused by intoxication. Phenobarbital is still the first-choice drug for the therapy of neonatal seizures.

The purpose of the benzodiazepine bolus is to stop GCSE, the purpose of the infusion of the second-choice drug is to prevent the recurrence of GCSE. If the termination of the GCSE benzodiazepines bolus occurs simultaneously with the infusion of the second-choice drug, it is appropriate to continue maintenance therapy with the second-choice drug.

After GCSE, most children regain consciousness within 20 to 30 min of the end of convulsions. The two most common reasons for delayed recovery are the sedative effect of ASM and non-convulsive SE, which is not possible to distinguish without EEG. An EEG should be performed within a few hours at the latest if the patient remains non-contact.

Note: Knowledge of previous treatment responses and knowledge of current pharmacological history may be guides to GCSE therapy. For example, if epileptic seizures are known to have not responded to phenytoin, the use of IV valproic acid or levetiracetam is offered.

On the other hand, if we know that the patient was compensated, for example, during treatment with valproic acid and that the patient missed one or more doses (or we assume a decrease in the level of valproic acid, for example, as a result of repeated vomiting), it is rational to intravenously apply this ASM.

## 4. Refractory GCSE (>40 Min)

If the GCSE is not stopped by benzodiazepine boluses with a simultaneous infusion of a non-benzodiazepine antiepileptic drug, i.e., at 30 to 40 min after development, convulsions are usually gradually mitigated and the GCSE tends to resemble subtle SE (see above). In this situation, it is usually necessary to start a general anesthesia with intubation and ventilation.

In the case of a less dramatic situation—i.e., if oxygenation and/or ventilation is not significantly compromised—it is possible to start an infusion of the second choice medication (e.g., if the GCSE persists after the phenytoin infusion, proceed to valproic acid or levetiracetam therapy) and to consider general anesthesia only if this second-choice second drug fails.

To introduce and maintain general anesthesia, we have a choice of one of three intravenous total anesthetics: midazolam, thiopental, and propofol (midazolam is, more precisely, a benzodiazepine hypnotic). Pentobarbital is not available in the Czech Republic. In children, the first-choice anesthetic is midazolam, less often thiopental. The use of propofol is associated with the risk of developing propofol infusion syndrome (see below).

However, propofol has a significant advantage over midazolam and especially thiopental: a significantly shorter duration of action and thus a significantly faster recovery from general anesthesia. In contrast, midazolam, compared to propofol and thiopental, does not cause such a pronounced depression of circulation (hypotension).

Midazolam 0.2 mg/kg (or additional boluses of 0.1–0.2 mg/kg until effect) –> 0.1–2 mg/kg/h (https://www.uptodate.com/contents/management-of-convulsive-status-epilepticus-in-children) (accessed on 23 November 2021).

Thiopental 3–5 mg/kg (or additional boluses of 1–2 mg/kg until the effect is achieved) –> 3–5 mg/kg/h [12].

Propofol 1–2 mg/kg (or additional boluses of 1–2 mg/kg until the effect is achieved) –> 2–4 mg/kg/h [12]. If the desired effect is not achieved at this dosage (burst suppression—see below), combine with midazolam (do not increase the dose).

After the introduction of general anesthesia, it is necessary to achieve the optimal depth of anesthesia, which is not possible without continuous EEG monitoring. Too shallow general anesthesia leads to a recurrence of GCSE after the reduction and discontinuation of the general anesthetic. On the other hand, too deep anesthesia causes undesirable attenuation of vital functions, while the main danger is alteration of circulation or hypotension (breathing can be fully replaced by controlled ventilation).

The goal of therapy is general anesthesia in which there is a burst suppression pattern in the EEG (see Figure 1). Adequate general anesthesia should be maintained—under continuous EEG monitoring (the target dosage of the total anesthetic may vary over time)—for 24 to 48 h. Simultaneously with general anesthesia, it is appropriate to continue maintenance therapy with second choice ASM.

In treated people with epilepsy, there is an effort to achieve full levels (possibly by increasing the doses) of the ASM taken. ASM that are not available in injectable form are administered enterally through a nasogastric probe. The purpose of these procedures is to prevent the recurrence of GCSE after the reduction and discontinuation of the general anesthetic.

General anesthesia termination should take place gradually. Terminating general anesthesia too quickly carries a risk of GCSE recurrence. For example, the dosage of propofol is reduced by 5% per hour. Midazolam and thiopental in particular may be discontinued more rapidly due to the longer duration of action. Continuous EEG monitoring is particularly important here because the recurrence of electrographic seizures can precede the recurrence of convulsions (See Figure 2).

## 5. Propofol Infusion Syndrome

Propofol infusion syndrome (PRIS) is a rare but potentially fatal syndrome with high mortality that occurs in critically ill children following a long-lasting continuous infusion of propofol at high doses. The pathophysiological basis is a disorder of the utilization of free fatty acids at the mitochondrial level, leading to a failure of cellular energy metabolism [13].

There is also a blockade of beta adrenoreceptors and calcium channels in the myocardium, which leads to a rather sudden refractory bradycardia with a tendency to progression to asystole [12,13].

At the same time, at least one of the following symptoms may be present:►Rhabdomyolysis with myoglobinuria, which secondarily leads to acute renal failure with hyperkalemia;►Hepatomegaly or hepatic steatosis with elevation of liver enzymes;►Hyperlipidemia or hypertriglyceridemia;►Severe metabolic or lactate acidosis.

Early signs of PRIS development are otherwise unexplained lactic acidosis, creatine kinase elevation, myoglobin, and hypertriglyceridemia. In particular, the development of etiologically unclear lactic acidosis early after the start of propofol infusion may predict the development of PRIS.

There is a risk of PRIS development if propofol is administered continuously for more than 48 h at a dose of more than 4 mg/kg/h. Risk factors for PRIS include concomitant treatment with corticosteroids and/or catecholamines, and sepsis and possibly septic shock [12,13].

PRIS therapy: When PRIS development is suspected, it is necessary to immediately stop the infusion of propofol. Symptomatic therapy is also necessary: temporary cardiostimulation, inotropic and vasopressor support of circulation, hemodialysis, etc. Despite adequate therapy, the mortality of PRIS is unacceptably high.

Prevention of PRIS: In patients at risk, consider the use of another general anesthetic (midazolam, thiopental). When deciding to use propofol, the dose should not exceed 4 mg/kg/h, especially when more than 48 h of propofol are needed.

## 6. Super-Refractory GCSE (Lasting More Than 24 h)

GCSE, which—despite 24 to 48 h of adequate general anaesthesia– relapses when general anesthetics are discontinued is more often associated with a poor prognosis. In these cases, the patient should be “returned” to the burst-suppression pattern in EEG for another 24 to 48 h [14].

In this situation, it is justified to use another anesthetic for further general anesthesia—e.g., thiopental or propofol in case of failure of midazolam (both of these anesthetics can also be combined with midazolam). At the same time, one or more of the following options may be considered:►Intravenous ketamine anesthetic at a dosage of 1–5 mg/kg/h (usual starting dose is a bolus of 1–2 mg/kg) [11,15], ketamine can be combined with all three listed intravenous general anesthetics;►High doses of magnesium sulphuric intravenously—possibly up to target mg levels of 3.5 mmol/L [11,15];►High doses of pyridoxine intravenously (30 mg per kg followed by up to 300 mg per day) in cases where pyridoxine-dependent epilepsy cannot be excluded [11,15];►Pulse doses of methylprednisolone + IVIG +—plasmapheresis, if the autoimmune etiology of GCSE cannot be excluded [11,15]; e.g., if NMDA-receptor encephalitis is suspected; when this encephalitis is detected, the use of cyclophosphamide or rituximab may be considered—in case of failure of the immunotherapy mentioned above [15];►Intravenous local anesthetic lidocaine (2 mg/kg followed by 2 mg/kg/h) [11];►Inhaled anesthetic drugs—isoflurane or desflurane (not sevoflurane) [11]; although an effective solution when achieving adequate levels of inhalation anesthetic, there is a tendency to recurrence of GCSE during withdrawal;►Therapeutic hypothermia 32–35 °C [11,15];►Rational cooling to a target central temperature of 34–35 °C (at temperatures <34 °C there is a risk of serious side effects);►Ketogenic diet (in the ketogenic diet, the use of propofol is contraindicated) [11,15];►Epileptosurgical resection of a clearly proven epileptogenic focus in an operable area of the brain [11];►Implantation of vagal nerve stimulation– in cases where epileptogenic focus cannot be clearly demonstrated [11];►Various forms of stimulation therapy (transcranial magnetic stimulation, deep brain stimulus, electroconvulsive therapy) and liquor drainage [11] must be considered experimental;►There are isolated references to the successful therapy of super-refractory SE with the neurosteroid allopregnanolone [16] and successful therapy of FIRES with a modified interleukin 1 receptor antagonist—anakinra [17,18].

## 7. Conclusions

GCSE is a critical condition, with the risk of irreversible neurological changes, recurrence of seizures, and a possible fatal outcome. Treatment must be initiated in the impending GCSE phase, i.e., within five minutes of the onset of a generalized seizure.

## 8. Key Points

(1)Generalized convulsive status epilepticus is emergency condition in children with high morbidity and mortality and potentially irreversible brain damage;(2)Early initiation and adequate treatment is crucial for good patient outcome;(3)Management of generalized convulsive status epilepticus depends on the stages of GCSE: early GCSE (5 to 20 min); developed GCSE (lasting 20 to 40 min); refractory GCSE (lasting more than 40 min); super-refractory GCSE (lasting more than 24 h);(4)Treatment must be initiated in the impending GCSE phase, within five minutes of the onset of a generalized convulsive seizure.

## Figures and Tables

**Figure 1 children-09-01586-f001:**
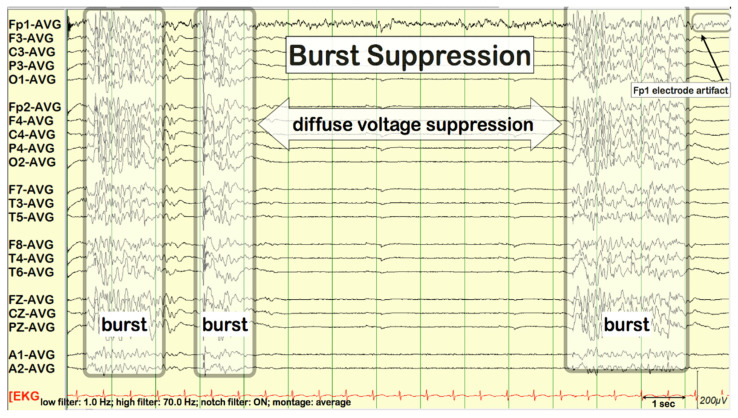
Burst-suppression pattern.

**Figure 2 children-09-01586-f002:**
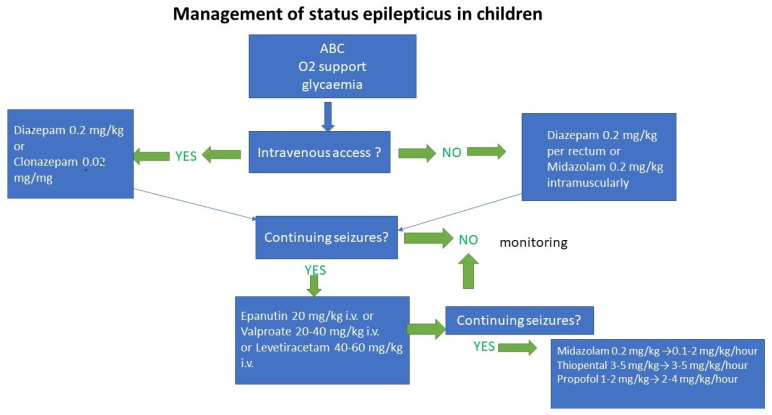
Management of status epilepticus in children.

**Table 1 children-09-01586-t001:** Diagnosis of etiology.

Population of Patients	Examinations
In all patients	Blood count and biochemical examination CT or MRI of the brain * EEG ** Toxicology screening in indicated cases
In patients with epilepsy	Levels of antiepileptic drugs according to pharmacological history
In febrile patients	► Blood count + differential ► CRP, procalcitonin ► CSF examination ***

* Brain imaging (MR or CT scan of the brain) is always indicated in cases when neuroimaging has not yet been performed and if there is no presumed restoration of the patient’s consciousness after the end of the seizure. For each patient who has undergone CSE, MRI of the brain should be performed in the future, although CT scans of the brain may be sufficient in the acute phase. ** In the initial phase of SE, EEG is used primarily for the differential diagnosis of non-epileptic psychogenic seizures. In the refractory and super-refractory CSE phase, the role of EEG in monitoring the depth of general anesthesia is irreplaceable. *** If meningoencephalitis is suspected, a lumbar puncture should be performed to examine the cerebrospinal fluid (always after a prior brain imaging that excludes intracranial hypertension). Brain imaging and lumbar puncture—if indicated—should be performed as soon as convulsions are controlled with conventional antiepileptic agents. In the case of refractory CSE, this should be in the first hours of general anesthesia.

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
