# Peer review of "Current Management of Generalized Convulsive Status Epilepticus in Children"

_children, 2022, doi:10.3390/children9101586_

Round 1

Reviewer 1 Report

The figure 1 is not clear, please revised the more clear one.

Please outline the key points ( maybe 4-5 ponts) for this manuscript to attract the readers. 

Please provide the table of  drugs dose/notes/side effect mentioned in the manuscript. 

Author Response

Reviewer 1

  • The figure 1 is not clear, please revised the more clear one.

Revised.

  • Please outline the key points ( maybe 4-5 points) for this manuscript to attract the readers. 

Added.

  • Please provide the table of drugs dose/notes/side effect mentioned in the manuscript. 

Added in algorithm– see Figure 2.

Reviewer 2 Report

This is an interesting review article, which should be accepted after a minor revision.

According to the referee's viewpoint, only one point needs to be addressed:

1) The text should be supplied by a graphical scheme briefly the showing algorithms of threatment of the GCSE in children. If it is possible, such a scheme can be shown on a graphical abstract.

Author Response

Reviewer 2

  • The text should be supplied by a graphical scheme briefly the showing algorithms of treatment of the GCSE in children. If it is possible, such a scheme can be shown on a graphical abstract.

Added- see Fig. 2 in main text.
